# Reduced Humoral and Cellular Immune Response to Primary COVID-19 mRNA Vaccination in Kidney Transplanted Children Aged 5–11 Years

**DOI:** 10.3390/v15071553

**Published:** 2023-07-14

**Authors:** Jasmin K. Lalia, Raphael Schild, Marc Lütgehetmann, Gabor A. Dunay, Tilmann Kallinich, Robin Kobbe, Mona Massoud, Jun Oh, Leonora Pietzsch, Ulf Schulze-Sturm, Catharina Schuetz, Freya Sibbertsen, Fabian Speth, Sebastian Thieme, Mario Witkowski, Reinhard Berner, Ania C. Muntau, Søren W. Gersting, Nicole Toepfner, Julia Pagel, Kevin Paul

**Affiliations:** 1University Children’s Research, UCR@Kinder-UKE, University Medical Center Hamburg-Eppendorf, Martinistr. 52, 20246 Hamburg, Germany; jasmin-kaur.lalia@stud.uke.uni-hamburg.de (J.K.L.); gadunay@protonmail.com (G.A.D.); freya.sibb@gmail.com (F.S.); gersting@uke.de (S.W.G.); 2University Children’s Hospital, University Medical Center Hamburg-Eppendorf, Martinistr. 52, 20246 Hamburg, Germany; r.schild@uke.de (R.S.); j.oh@uke.de (J.O.); u.schulze-sturm@uke.de (U.S.-S.); f.speth@uke.de (F.S.); muntau@uke.de (A.C.M.); ju.pagel@uke.de (J.P.); 3Institute of Medical Microbiology, Virology and Hygiene, University Medical Center Hamburg-Eppendorf, Martinistr. 52, 20246 Hamburg, Germany; mluetgehetmann@uke.de; 4German Center for Infection Research (DZIF), Partner Site Hamburg-Lübeck-Borstel-Riems, Inhoffenstr. 7, 38124 Brauschweig, Germany; 5Department of Pediatric Respiratory Medicine, Immunology and Critical Care Medicine, Charité University Medicine Berlin, Charitéplatz 1, 10117 Berlin, Germany; tilmann.kallinich@charite.de; 6Institute for Infection Research and Vaccine Development (IIRVD), University Medical Center Hamburg-Eppendorf, Martinistr. 52, 20246 Hamburg, Germany; r.kobbe@uke.de; 7Department of Infectious Disease Epidemiology, Bernhard-Nocht-Institute for Tropical Medicine, Bernhard-Nocht-Straße 74, 20359 Hamburg, Germany; 8Therapeutic Gene Regulation, Deutsches Rheuma-Forschungszentrum (DRFZ), An Institute of the Leibniz Association, Charitéplatz 1, 10117 Berlin, Germany; mona.massoud@drfz.de; 9Department of Pediatrics, Faculty of Medicine and University Hospital Carl Gustav Carus, Technische Universität Dresden, Fetscherstraße 74, 01307 Dresden, Germany; leonora.pietzsch@ukdd.de (L.P.); catharina.schuetz@ukdd.de (C.S.); sebastian.thieme@ukdd.de (S.T.); reinhard.berner@ukdd.de (R.B.); nicole.toepfner@ukdd.de (N.T.); 10Institute of Microbiology, Infectious Diseases and Immunology, Laboratory of Innate Immunity, Charité University Medicine Berlin, Charitéplatz 1, 10117 Berlin, Germany; mario.witkowski@charite.de; 11Mucosal and Developmental Immunology, Deutsches Rheuma-Forschungszentrum (DRFZ), An Institute of the Leibniz Association, Charitéplatz 1, 10117 Berlin, Germany; 12Division of Pediatric Stem Cell Transplantation, Immunology and Rheumatology, University Medical Center Hamburg-Eppendorf, Martinistr. 52, 20246 Hamburg, Germany

**Keywords:** T cell, pediatric, SARS-CoV-2, solid organ transplant, glomerulonephritis

## Abstract

The situation of limited data concerning the response to COVID-19 mRNA vaccinations in immunocom-promised children hinders evidence-based recommendations. This prospective observational study investigated humoral and T cell responses after primary BNT162b2 vaccination in secondary immunocompromised and healthy children aged 5–11 years. Participants were categorized as: children after kidney transplantation (KTx, *n* = 9), proteinuric glomerulonephritis (GN, *n* = 4) and healthy children (controls, *n* = 8). Expression of activation-induced markers and cytokine secretion were determined to quantify the T cell response from PBMCs stimulated with peptide pools covering the spike glycoprotein of SARS-CoV-2 Wuhan Hu-1 and Omicron BA.5. Antibodies against SARS-CoV-2 spike receptor-binding domain were quantified in serum. Seroconversion was detected in 56% of KTx patients and in 100% of the GN patients and controls. Titer levels were significantly higher in GN patients and controls than in KTx patients. In Ktx patients, the humoral response increased after a third immunization. No differences in the frequency of antigen-specific CD4+ and CD8+ T cells between all groups were observed. T cells showed a predominant anti-viral capacity in their secreted cytokines; however, this capacity was reduced in KTx patients. This study provides missing evidence concerning the humoral and T cell response in immunocompromised children after COVID-19 vaccination.

## 1. Introduction

In late 2021, vaccination efforts against COVID-19 were expanded with the approval of mRNA vaccines for the pediatric population of children aged 5–11 years. With a reduced dosage of 10 µg instead of 30 µg BTN162b2 (COMIRNATY^®^ BIONTECH/Pfizer), phase II/III clinical trials have shown comparable immunization effects in adolescent populations to adults regarding neutralizing antibody production [1]. Additionally, a strong T cell response was demonstrated in healthy children aged 5–11 years after BTN162b2 vaccination [2]. Currently, primary, as well as booster vaccinations, are encouraged for immunosuppressed children [3,4]. However, as only limited data [5,6] on the immunological response to the vaccination exist in this age group, expert-based recommendations have displaced evidence-based recommendations.

While a higher COVID-19 morbidity rate was observed among adult and pediatric immunocompromised populations [7,8,9], disease severity, especially in pediatric transplant recipients, appears to be mild [10,11,12,13]. Nevertheless, SARS-CoV-2 infection resulted in higher levels of symptomatic disease and acute kidney injury in pediatric kidney transplant (KTx) recipients [14]. Additionally, children between 5–11 years were reported to have the highest risk of developing Multisystem Inflammatory Syndrome in Children after an infection with SARS-CoV-2 [15,16], which has been shown to be reduced upon COVID-19 vaccination [17]. In conclusion, there is a need for data enabling evidence-based specific recommendations for COVID-19 vaccination of immunocompromised children aged 5–11 years.

With this prospective observational study, we aimed to investigate whether protective humoral and T cell responses in immunocompromised pediatric patients aged 5–11 years could be obtained after primary vaccination with two doses of BNT162b2. As such, humoral and T cell responses were assessed in secondary immunocompromised children with either proteinuric glomerulonephritis (GN) receiving only calcineurin inhibitors (CNI) or pediatric patients after KTx and healthy controls. These cohorts allowed us to compare various levels of immunosuppressive medication, as healthy children received no immunosuppression, GN patients received light immunosuppression, and KTx patients received strong immunosuppression. The results of this study will provide further evidence for current and future vaccine recommendations.

## 2. Materials and Methods

### 2.1. Study Cohort and Ethics

Patients were recruited from the cohort of the German ‘PaedVacCOVID’ study (initiated by the University Hospital Carl Gustav Carus Dresden) at the University Medical Center, Eppendorf, Hamburg from January to August 2022 during their routine examinations based on the following criteria: patients post KTx or patients with GN. Parents and healthy children of the “Prospektive minimal-interventionelle Beobachtungsstudie an Personen, die mit einem zugelassenen Impfstoff gegen COVID-19 geimpft wurden,” (Prospective minimal-intervention observational study of subjects vaccinated against COVID-19 with an approved vaccine) were asked to voluntarily participate at the vaccination center at University Medical Center, Eppendorf, Hamburg, before their first vaccine dose and after their second dose. Blood samples were obtained at pre-defined time points: before the first vaccination (T0), 3–10 weeks after the second vaccine dose (T2), and at least three weeks after the third vaccine dose or natural infection (T3). Participants were only included in the study if a sample at T2 was available. During the study period, children in Hamburg, Germany underwent regular, biweekly nasal swab SARS-CoV-2 antigen testing in school or day care. Study exclusion criteria were defined as follows: a past medical history of SARS-CoV-2 infection, or children who had tested positive for SARS-CoV-2 nucleocapsid IgG/M/A before completing the primary vaccination (before T2). At T3, SARS-CoV-2 nucleocapsid IgG/M/A status was assessed. If a natural infection was acquired and a third vaccine dose was applied, patients were also excluded from timepoint T3 analysis.

The recruitment of immunocompromised children as part of the ‘PaedVacCovid’ study as well as of healthy controls within the study “Prospektive minimal-interventionelle Beobachtungsstudie an Personen, die mit einem zugelassenen Impfstoff gegen COVID-19 geimpft wurden” was approved by the ethics committee of the Ärztekammer Hamburg (Ref: 2022-200298-BO-bet and Ref: 2020-10376_4-BO-ff, respectively)

### 2.2. Sample Processing

Serum was obtained from whole blood using centrifugation and stored as aliquots in liquid nitrogen until analysis.

PBMCs were isolated within eight hours from Lithium Heparin blood (1–10 mL) by a density gradient centrifugation using SepMate tubes^®^ and Lymphoprep^®^ (StemCell, Cologne, Germany) according to the manufacturer’s recommendations. PBMCs were cryopreserved in freezing medium containing 30% RMPI (Gibco, Life Technologies Limited, Paisley, UK), 20% DMSO (Chemsolute, TH.GEYER GmbH & Co. KG, Renningen, Germany), and 50% FCS (Capricorn, Ebsdorfergrund, Germany) and stored in liquid nitrogen.

### 2.3. Serum Antibody Measurements

To quantify the humoral response after primary vaccination against COVID-19, SARS-CoV-2 spike-specific antibodies directed against the receptor-binding domain (RBD) were measured in serum using the Elecsys^®^ Anti-SARS-CoV-2 Ig assay (Roche, Mannheim, Germany), at a cut off of 0.8 U/mL.

Antibodies directed against the viral nucleocapsid (Roche Elecsys^®^ Anti-SARS-CoV-2 Ig assay, cut off 1.0 U/mL) were assessed to identify whether a prior SARS-CoV-2 infection had taken place before the second vaccination dose was administered.

Analyses were performed according to in vitro diagnostics criteria on the cobas e411 system (Roche) as recommended by the manufacturer.

### 2.4. PBMC Peptide Stimulation

PBMC peptide stimulation was conducted as described previously [18], with slight modifications. PBMCs were thawed in a prewarmed water bath for one minute at 37 °C and diluted in 30 mL warm RPMI (Gibco). Cells were incubated with RPMI, 5% human serum (HS) (human serum AB male bio&sell), and 50 U/mL Benzonase (Benzonase^®^ Sigma-Aldrich Chemie GmbH, Taufkirchen, Germany) for one hour at 37 °C 5% CO_2_. After washing with RMPI and 5% HS, PBMCs were left to rest for 12 h at 37 °C, 5% CO_2_ in 1 mL RMPI and 5% HS. Following a washing step, cells were transferred to four wells of a 96-well U-bottom plate containing 200 µL RPMI and 5% HS. Antigen-specific T cells were stimulated for 24 h at 37 °C and 5% CO_2_ by adding pools of overlapping peptides, 15-mer by 10 amino acids at 1 µg per peptide/mL, covering either the spike protein of SARS-CoV-2 original strain Wuhan-Hu-1 (Spike-OS) or SARS-CoV-2 Omicron BA.5 (Spike-BA5) variant (Peptides & elephants GmbH, Henningsdorf, Germany). DMSO (Chemsolute) in an equimolar amount was used to dissolve peptide pools (0.2%) served as negative control. PHA-L (Invitrogen^TM^ eBioscience^TM^, Thermo Fisher Scientific, Carlsbad, CA, USA) 1 µg/mL was used as a positive control. After stimulation, cell-free culture supernatant was removed from each well and stored at −20 °C for future cytokine analysis. PBMCs were washed in 2 mL PBS to stop the stimulation and allow for flow cytometry analysis. Antigen-specific T cells were identified by their expression of activation-induced markers (AIM), namely OX40 and CD69 (CD4+ T cells) or CD137 and CD69 (CD8+ T cells).

### 2.5. Flow Cytometry

After washing PBMCs with PBS to stop stimulation, they were incubated for 15 min with a Near-IR Dead Cell Stain Kit (Invitrogen). Without washing, cells were stained using an antibody cocktail (Appendix A) for 20 min at room temperature (RT). After washing with PBS, cells were fixed with 1% PFA (Morphisto GmbH, Offenbach am Main, Germany) for at least 1 h at 4 °C. Washed cells were then stored at 4 °C until flow analysis was performed within 8 h.

Rainbow Calibration Particles (BD Sphero, BD Biosciences, Heidelberg, Germany) were used to ensure the consistency of fluorescence intensities between experiments. Anti-Mouse Ig and k/Negative Control Compensation Particles Set (BD Biosciences) were used for antibody compensation, while ArC Amine Reactive Compensation Beads (Invitrogen) were used to compensate for the Dead Cell Stain Kit as described before.

Flow analysis was performed with a BD FACSymphony A3 flow cytometer in the Cytometry and Cell Sorting Core Unit at the University Medical Center, Eppendorf, Hamburg.

The gating strategy for the AIM+ T cells is depicted in detail in Appendix A. The Protocol was previously described in [18].

### 2.6. Multiplex Detection of Cytokines

Secreted cytokines in the cell culture supernatant were quantified using a cytokine bead array suitable to detect IL-2, IL-4, IL-10, IL-6, IL-17A, TNF-α, sFas, sFasL, IFN-γ, Granzyme A, Granzyme B, Perforin and Granulysin (LEGENDplex^®^ Human CD8/NK Panel 13-plex, BioLegend, San Diego, CA, USA). Experiments were performed according to the manufacturer’s instructions and as described previously [18,19].

### 2.7. Data Analysis and Statistics

FlowJo version 10 and R 4.0.5 (packages: tidyverse, rstatix, splines, emmeans, kableExtra, magrittr, heatmaply) were used to perform data analysis and process graphs, and statistics. Data of antibody titers, cell frequencies, the stimulation index of the AIM assay and fold increase of secreted cytokines showed highly skewed distributions. Therefore, log transformation of these data was applied for further statistical analysis.

Groups were compared using one-way ANOVA with post hoc pairwise *t*-tests. No adjustment for multiple testing was necessary for pairwise comparisons following the closed test principle. For paired samples, paired *t*-tests were used.

Analysis of AIM+ cells was conducted as described previously [19,20,21]. To compare cell frequencies between groups, a stimulation index (SI) of each individual’s T cell response was calculated. In CD4+ T cells, the frequency of AIM+ cells (OX40+CD69+) after peptide stimulation was divided by the frequency of the same cell subset in the negative control (DMSO). The stimulation index in CD8+ T cells was calculated with the frequency of CD137+ CD69+ cells accordingly.

A stimulation index > 3 in CD4+ T cells and > 2 in CD8+ T cells was defined as a response. In preliminary experiments, these thresholds assured a distinction between unspecific responses while maintaining assay sensitivity.

If no AIM+ cells were detected, those zero values, which would interfere with data processing, were replaced by half of the lowest measured cell frequency of the respective cell population among all samples. Accordingly, half of the detection limit was set to cytokine concentrations which were below the limit of detection defined by the manufacturer.

Samples with fewer than 1000 CD4+ or CD8+ T cells were excluded from AIM as well as cytokine analyses.

In the case of low cell counts, if available, another aliquot of patient PBMC was analyzed. To achieve the required cell number, fcs files were concatenated. However, these samples were still excluded from cytokine analysis.

Statistical significance was set at *p* < 0.05.

## 3. Results

### 3.1. Cohort Characteristics

A total of 180 participants were recruited for this study. Children who acquired a natural infection prior to T2 were excluded. Additionally, the recruited patients were characterized by a high diversity in underlying diseases and schemes of immunosuppression. Thus, if a statistically evaluable group size was not reached for the respective primary disease subgroups, those patients were also excluded from further analysis.

The final cohort consisted of a total of 21 participants. Nine KTx children were enlisted. Two of these children experienced a natural infection following their second COVID-19 vaccine dose, and one was given a third dose. The mean time after transplantation was 5.5 years. On average, patients with KTx were 7.7 years old and four (44%) were female. One patient only received CNI after KTx, while seven patients received CNI plus mycophenolate mofetil (MMF), and one patient received CNI and a mTOR-inhibitor (Table 1).

Four children with GN were recruited at T2, all diagnosed with nephrotic syndrome. The mean age of GN patients was 9.25 years and two (50%) were female. All GN patients received CNI as a single immunosuppressive treatment.

The final analysis included eight healthy controls, with a mean age of 7.6 years. In total, five individuals were female (62.5%).

There is a gradation of immunosuppressive medication in these cohorts. As a result, the various degrees of immunosuppression can be compared. All study participants received vaccine doses of 10 µg BNT162b2.

### 3.2. Humoral Immune Response

To identify the humoral immune response, we assessed SARS-CoV-2 spike RBD IgG/M/A titers of all participants at each pre-defined timepoint.

Prior to the first vaccine dose, antibodies against spike RBD were not detected in healthy controls. Following the primary vaccination, a seroconversion and a significant increase in antibody titers in all healthy participants (8 of 8, 100%) were observed. Moreover, a seroconversion was seen in 100% (4 of 4) of GN patients (Figure 1).

In contrast, only five out of nine (56%) KTx patients seroconverted after the primary vaccination (T2). In line with this, overall lower antibody titers were detected in KTx patients (Figure 1). However, it should be noted that three KTx patients demonstrated an increase in spike RBD antibody titers after receiving a third vaccine dose or contracting a natural infection (Figure 1). Notably, a seroconversion was seen at T3 in one KTx patient, who did not show a humoral response at T2 after experiencing a natural infection between T2 and T3.

### 3.3. CD4+ T Cell Response

Specific responses to the Wuhan-Hu-1 (Spike-OS) and Omicron BA.5 (Spike-BA5) strain were detected based on the expression of activation-induced markers CD69 and OX40 in CD4+ T cells (Figure 2). According to preliminary experiments, to be considered as responders, an SI of 3 or higher was required.

In healthy children, a significant increase in SI upon Spike-OS stimulation between T0 and T2 was found. The increase in specific CD4+ T cell responses to Spike-BA5, however, did not reach a level of significance (Figure 3a,b).

In healthy controls, six out of eight (75%) had a positive CD4+ T cell response upon peptide stimulation to Spike-OS after two vaccine doses (Figure 3a). The Spike-BA5 peptide pool elicited a response in five out of eight (63%). In one healthy participant, antigen-specific T cells to both peptide pools were detected before the first vaccine dose, although the serology was negative, possibly indicating preexisting cross-reactive T cells.

All children with GN responded positively to the Spike-OS stimulation, with 2 out of 3 (67%) further responding to the Spike-BA5 stimulation at T2. Spike-OS responses were seen in 67% (four of six) of KTx patients, and Spike-BA5 responses in five of six (83%) KTx patients. A difference in CD4+ T cell response strength after two vaccine doses, defined by SI, was not seen between the cohorts. This underscores the capability of the investigated immunocompromised patients to develop a relevant specific CD4+ T cell response after BNT162b2 vaccination, as the responses to Spike-OS were similar in both GN and KTx patients to those of healthy children.

The SARS-CoV-2 Wuhan-Hu-1 strain-based vaccine also leads to the generation of Omicron BA.5 cross reactive CD4+ T cells in healthy and immunocompromised children (Figure 3b). No difference in cross-reactive T cell response, defined by SI, could be detected between the three cohorts.

Additionally, the memory phenotype of specific T cells was analyzed by their expression of CD45RA and CD27 (Appendix A). Compared to total CD4+ T cells, central and effector memory T cells were especially enriched in AIM+ CD4+ T cells (Figure 3c,d), showing that immunization elicited the development of memory CD4+ T cells that are able to proliferate upon antigen encounter and home in secondary lymphatic tissues [22].

### 3.4. CD8+ T Cell Response

CD8+ T cell responses were defined based on the expression of the AIM markers CD137 and CD69. The principle was similar to CD4+ T cells; a specific response was defined as an SI > 2 (Figure 2). In preliminary experiments in CD8+ T cells, an SI > 2 was sufficient to separate specific responses from unspecific activation.

In two out of eight (25%) healthy children, specific CD8+ T cells to Spike-OS and Spike-BA5 could be detected before vaccination. After receiving the second vaccine dose, the response rates to Spike-OS were 63% (5 of 8) for controls, 100% (3 of 3) for GN patients, and 50% (4 of 8) for KTx patients. The specific CD8+ T cell response was similar to Spike-BA5, namely: 63% (5 of 8) for controls, 100% (3 of 3) for GN, and 63% (5 of 8) for KTx in the same patients.

Despite the increase in response rate, the rise in SI between T0 and T2 in healthy controls was not significant in this small cohort (*p* = 0.075 Spike-OS; *p* = 0.079 Spike-BA5) (Figure 4a,b). When comparing SI, no differences in specific CD8+ T cell responses towards peptides derived from Spike-OS and Spike-BA5 were observed between the cohorts (Figure 4a,b).

These data indicate that primary vaccination moderately enhances the development of a SARS-CoV-2-specific CD8+ T cell response to the spike proteins of the Wuhan-Hu-1 and Omicron BA.5 strains. No significant difference in CD8+ T cell response, defined by SI, between healthy and immunocompromised patients could be seen, indicating an inferior effect of vaccination on the CD8+ T cell response, compared to the CD4+ T cell response.

### 3.5. Cytokine Response

To further characterize the phenotype and antiviral capacity of vaccine-elicited T cell responses, thirteen different cytokines in the cell culture supernatant after 24 h of peptide stimulation were quantified. Cytokine responses were calculated as fold increase after peptide stimulation over the respective DMSO negative control, to compare responses between groups and to correct for varying numbers of available stimulated PBMCs and relative frequency of T cells between samples.

Upon peptide stimulation, there was a significant increase in secreted IL-2, IL-10, IFN-γ, Granzyme B (Spike-OS and Spike-BA5) and additionally Granzyme A (Spike-BA5 only) detected in healthy controls when comparing corresponding samples from the T0 and T2 time point (Figure 5a–i). Thus, secretion of those cytokines upon specific T cell stimulation was demonstrated to be specifically influenced by primary vaccination. For the remaining cytokines, no difference between T0 and T2 was observed in healthy controls; therefore, detected secretion of those cytokines during peptide stimulation was rather unspecific.

Analysis of the cytokine response of IL-2, IL-10, IFN-γ, Granzyme A, and Granzyme B following second vaccination between cohorts showed that patients after KTx did secrete IL-10 and IFN-γ after Spike-OS stimulation and IL-10, IFN-γ, and Granzyme B after Spike-BA5 stimulation to a lower extent than in healthy controls. In GN patients, no difference in the amount of secreted IL-2, IL-10, IFN-γ, Granzyme A and Granzyme B compared to healthy controls could be detected.

In summary, the mRNA-based COVID-19 vaccine elicited a phenotypically diverse T cell response of a predominantly antiviral quality (IFN-γ, Granzyme) in both healthy and immunocompromised individuals. Furthermore, a regulatory response was seen with IL-10 production in all groups. The cytokine response was lower in Ktx patients but not in GN patients.

## 4. Discussion

In this prospective observational study, we investigated the humoral and specific T cell response after a primary BNT162b2 COVID-19 vaccination in immunocompromised pediatric patients with kidney disease aged 5–11 years in comparison with healthy controls.

In line with previous reports, a strong humoral response, with a 100% seroconversion rate, was demonstrated in healthy individuals after primary vaccination [1,2,23].

While titer levels after the second vaccination in GN patients receiving CNI did reach same levels as in healthy controls, the humoral response in investigated KTx patients was markedly decreased, and the overall seroconversion rate was 56%. Previously, a humoral response was reported in 62% of Ktx patients [6]. A reduced capability of primary vaccination to elicit a humoral response was also seen in adult and adolescent patients after solid organ transplantation [24,25,26]. Notably, in adolescent KTx patients aged 12–18 as well as in adult KTx patients, even lower seroconversion rates of 24% [26] and 4–48% [27] were reported. As such, these data suggest that KTx patients aged 5–11 years would exhibit a comparable, if not slightly stronger, humoral response after primary vaccination against COVID-19 with 10 µg BNT162b2.

In accordance with previous studies [28,29], our data also indicated that administration of multiple immunosuppressants, especially in combination with MMF, dampens the vaccine response. Further, our data underscore the reported beneficial effect of a third immunization on humoral response in Ktx patients [5,30,31]. An increase in antibody titer levels, as well as seroconversion in one additional patient, could be observed after the third immunization due to infection or vaccination in our KTx cohort.

The development of a robust T cell response is an important indicator to determine COVID-19 vaccine responses. Especially in immunocompromised patients with low humoral response rates, elicited specific T cells might still protect from severe COVID-19 disease courses [32].

Therefore, specific CD4+ and CD8+ T cell responses upon BNT162b2 vaccination to the vaccine Wuhan-Hu-1 as well as to the later evolved Omicron BA.5 variant were quantified.

While in unvaccinated healthy individuals, a preformed T cell immunity to Wuhan-Hu-1 spike protein was only apparent in 14% (CD4+) and 28% (CD8+), primary vaccination resulted in 75% (CD4+) and 63% (CD8+) responses in healthy participants. Previously, a response rate of 100% in healthy children of this age group was reported in an ELISPOT-based assay [2]. Taken together, these profound T cell responses stand in contrast with the reported reduced cellular response rates after SARS-CoV-2 infection in younger populations [18,33] and add a beneficial effect of vaccination over infection even for healthy children.

After vaccination, the amount of detected spike-specific CD4+ and CD8+ T cells obtained from GN and KTx patients did not differ from the healthy controls. Central and effector memory T cells were enriched in spike-specific CD4+ T cells, indicating a long-lasting T cell immunity upon primary vaccination. By additionally analyzing secreted cytokines in cell culture supernatant, a diverse and predominant antiviral T cell response was detected. Although quantitative T cell responses did not show a difference between healthy and immunocompromised children, in KTx patients, the capability of antigen-specific T cells to secrete specific cytokines appeared to be impaired upon antigen encounter. In conclusion, these results indicate a functionally reduced T cell response in KTx children to the BNT162b2 vaccination compared to healthy controls, which is in line with data from adult cohorts [29,34,35]. As seen in the humoral response, application of multiple immunosuppressants, especially MMF, appears to impair T cell effector function.

In addition, the cross reactivity of vaccine-induced T cells to the spike protein of the Omicron BA.5 variant was investigated. Similarly to Wuhan-Hu-1-specific T cells, no quantitative difference in detectable antigen-specific CD4+ and CD8+ T cells between patient cohorts was observed. This shows that primary vaccination with a Wuhan-Hu-1 strain-based vaccine elicits a relevant T cell response to a variant with significant antibody immune escape in healthy, as well as immunocompromised, pediatric patients. The capability of T cells to preserve a specific response despite mutations in the spike protein domain has been shown in healthy adults [36,37] and children [2,18] after SARS-CoV-2 infection or vaccination. However, as seen in Wuhan-Hu-1-specific T cells, the effector function, characterized by cytokine secretion upon stimulation, was reduced in the KTx cohort.

Hence, our results suggest that there is no need for a reduction in immunosuppression in pediatric KTx patients aged 5–11 years to achieve a stronger vaccine response, which was under discussion in adult populations [29,38]. With qualitatively reduced but quantitatively comparable T cell responses and comparably good humoral responses upon primary vaccination, together with reported mild disease progression upon infection, a reduction in immunosuppression is not justified considering possible organ rejection. As reported by Morgans et al. [5], the data also suggest the benefit of booster vaccinations to achieve higher antibody levels for KTx patients aged 5–11 years.

A clear limitation is the small sample sizes, which were mainly driven by SARS-CoV-2 immunization of potential candidates due to natural infection before vaccination. Our small sample size did not allow for multivariate analysis. We partially addressed this by comparing patients with single CNI-based immunosuppression to transplanted patients with mostly dual immunosuppression. Further, prior SARS-CoV-2 infection cannot absolutely be ruled out by testing for anti-nucleocapsid antibodies, particularly in the group of transplanted patients, since they seroconvert poorly. Nonetheless, from 2021 until the end of recruiting, children were regularly tested by rapid antigen testing in daycares and schools, which offered a rather reliable patient history. The strengths of this study are the clearly defined, albeit small cohort of KTx patients aged 5–11 years, and the possible support of these findings to clinical decision making regarding vaccination against SARS-CoV-2 as well as future mRNA vaccines currently under development [39]. Moreover, the study setting is somewhat unique in that baseline SARS-CoV-2 seronegative participants could be recruited, and this will become increasingly difficult in the age of high SARS-CoV-2 seroprevalence.

## 5. Conclusions

This study provides yet missing evidence for the effect of COVID-19 vaccinations in immunocompromised patients aged 5–11 years. The data demonstrate that a primary vaccination with 10 µg of BTN162b2 induces a relevant humoral along with a specific and Omicron BA.5 cross-reactive cellular response. SARS-CoV-2-specific immunity not only prevents from severe COVID-19 disease course [40], but may also provide an additional level of security for patient’s caregivers. This might result in a reduction of the self-imposed social distancing measures, which have been shown to increase depression and anxiety rates, as well as impaired social development [41,42,43]. Taken together with the reported low risk profile of the BTN162b2 vaccination in immunocompromised patients [44], the data underscores the beneficial effect of vaccination in this vulnerable population.

## Figures and Tables

**Figure 1 viruses-15-01553-f001:**
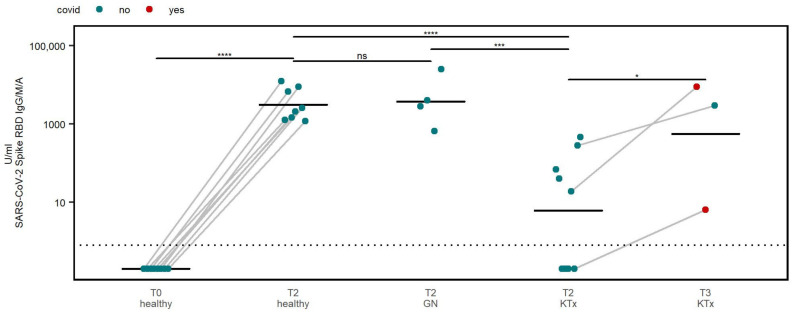
Serum samples were tested for SARS-CoV-2 spike-specific antibodies directed against the RBD and compared between cohorts at different timepoints. Dotted horizontal line indicates threshold to define seropositivity. Responses to different time points were compared by paired *t*-test. Inter group analyses were carried out by one-way ANOVA and post hoc pairwise *t*-tests. * *p* < 0.05, *** *p* < 0.001, **** *p* < 0.0001, ns—not significant.

**Figure 2 viruses-15-01553-f002:**
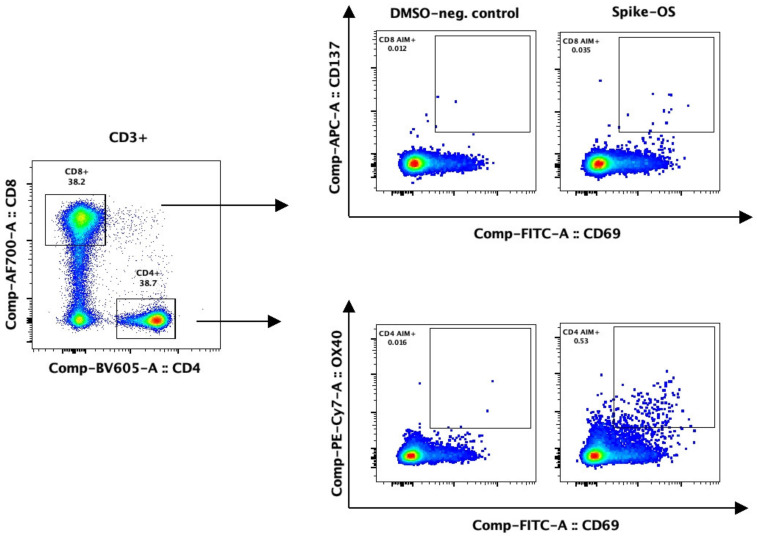
Flow cytometry results of a heathy control after two vaccinations, gated on living CD3+ lymphocytes, shown by density plots. After 12 h of rest after thawing, PBMCs were split evenly and stimulated for 24 h with DMSO (negative control), PHA-L (positive control), or SARS-CoV-2 peptide pools that covered the spike glycoprotein of the original strain Wuhan-Hu-1 (Spike-OS) or Omicron BA.5 variant (Spike-BA5). CD4 AIM+ T cells were detected by the expression of CD69 and OX40, and CD8+ T cells by the expression of CD69 and CD137. After the comparability of fluorescence intensity was confirmed by rainbow bead calibration, identical CD4 and CD8 AIM+ gates were set for all samples from all participants.

**Figure 3 viruses-15-01553-f003:**
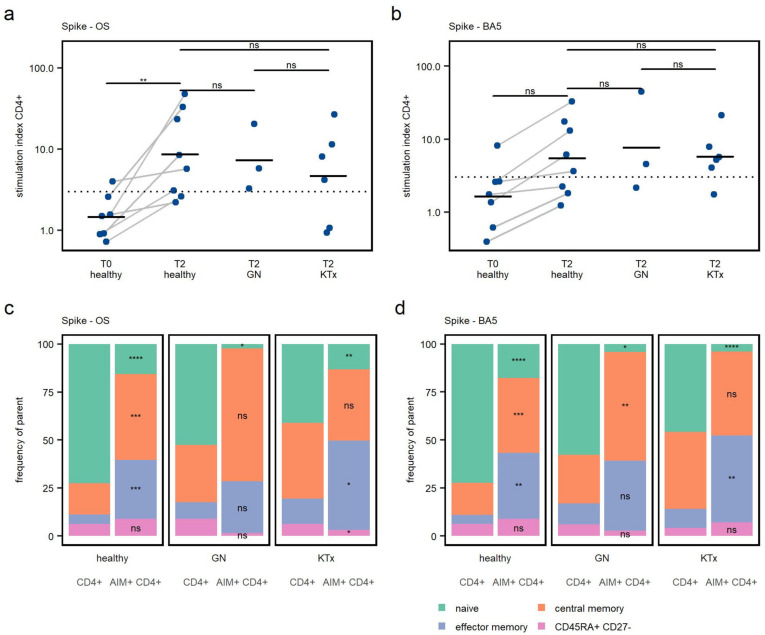
(**a**,**b**): Comparison of cohorts’ T cell responses to peptide stimulation. The frequency of AIM+ cells following peptide stimulation was divided by the frequency of AIM+ cells in the DMSO negative control to calculate the Stimulation Index (SI), which was used to measure the T cell response. Response was defined as SI > 3 (dashed line). Horizontal lines represent mean values. One-way ANOVA and post hoc pairwise *t*-tests were used to conduct statistical comparisons. (**c**,**d**): Memory phenotype of total CD4+ and AIM+ CD4+ T cells after peptide stimulation with Spike-OS and Spike-BA5. Mean values of the frequency within the respective parent population of study participants are displayed. Parent population is indicated at the bottom. Comparison of each T cell memory subset was performed between total CD4+ T cells and respective subset of AIM+ CD4+ T cells by using a two-sided paired *t*-test. * *p* < 0.05, ** *p* < 0.01, *** *p* < 0.001, *****p* < 0.0001, ns—not significant.

**Figure 4 viruses-15-01553-f004:**
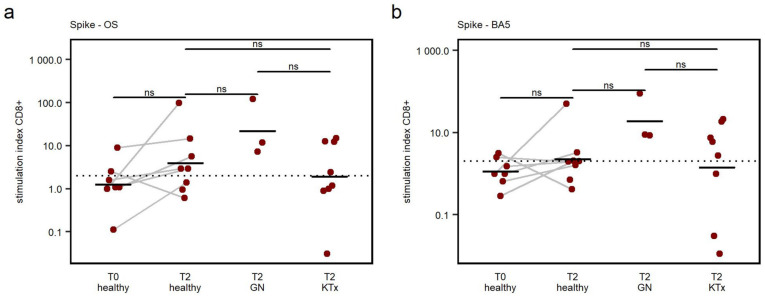
(**a**,**b**): T cell responses to peptide stimulation across cohorts. The Stimulation Index (SI), which was used to evaluate the T cell response, was calculated by dividing the frequency of AIM+ cells after peptide stimulation by the frequency of AIM+ cells in the DMSO negative control. An SI > 2 was used to define a response (dashed line). Mean values are represented by horizontal lines. Comparative analysis was carried out using post hoc pairwise *t*-tests and one-way ANOVA. ns—not significant.

**Figure 5 viruses-15-01553-f005:**
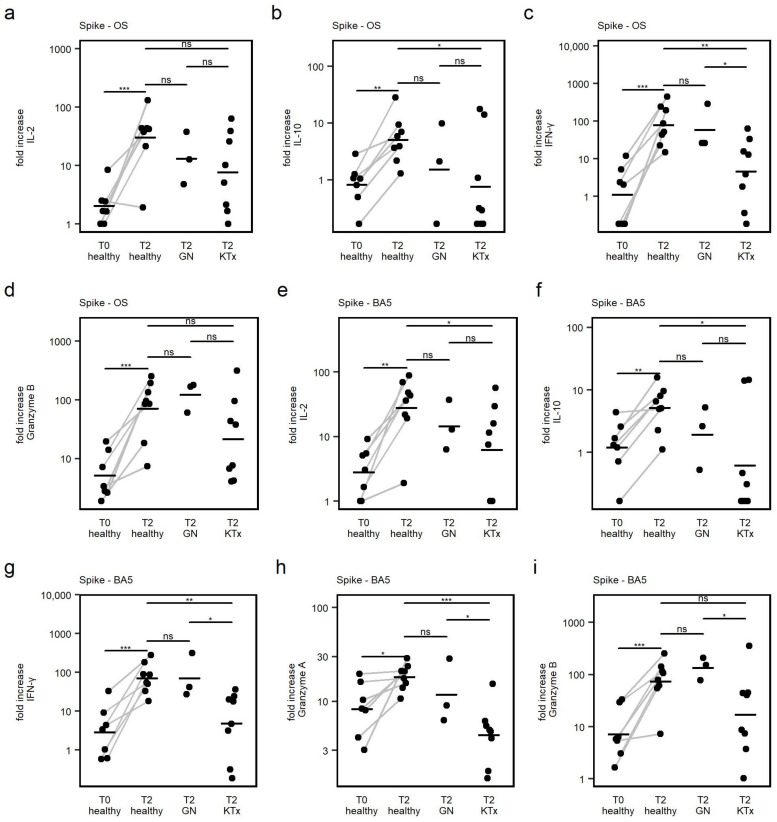
(**a**–**i**): After stimulation with Spike-OS and Spike-BA.5, the levels of 13 cytokines in cell culture supernatants were measured using a cytokine bead assay. The top of each graph shows which peptide pool was employed for stimulation. Cytokines showing a significant increase upon vaccination in healthy controls are displayed as examples. The comparison was carried out using the fold increase in cytokine concentration following peptide stimulation over cytokine concentration in corresponding DMSO treated samples in order to account for variations in the number of stimulated PBMCs and unspecific cytokine production. Horizontal lines represent mean values. P values were quantified using paired *t*-test (connected dots); intergroup analyses were carried out by post hoc pairwise *t*-tests and one-way ANOVA. * *p* < 0.05, ** *p* < 0.01, *** *p* < 0.001, ns—not significant.

**Table 1 viruses-15-01553-t001:** Number of patients (n) in each cohort according to disease. CNI: calcineurin inhibitors (cyclosporin A or tacrolimus) MMF: mycophenolate mofetil.

	Healthy	GN	KTx
Age, mean (range in years)	7.625 (5–11)	9.25 (7–11)	7.7 (5–11)
Sex (female, in %)	5 (62.5%)	2 (50%)	4 (44%)
**Medication**			
CNI		4	1
CNI + MMF			7
CNI + mTOR Inhibitor			1

## Data Availability

The raw data supporting the conclusions of this article will be made available by the authors, without undue reservation.

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
