# Peer review of "Reduced Humoral and Cellular Immune Response to Primary COVID-19 mRNA Vaccination in Kidney Transplanted Children Aged 5–11 Years"

_viruses, 2023, doi:10.3390/v15071553_

Round 1
Reviewer 1 Report
This prospective observational study investigated the humoral and T cell responses after primary BNT162b2 vaccination in secondary immunocompromised and healthy children aged 5 – 11 years. Overall, the topic is interesting and the manuscript is well-written. However, the case number is too limited to be presented as a full article. Therefore, I would recommended that the article could be shortened and accepted as a “Brief report”.
Author Response
Point to point response to Reviewer 1
This prospective observational study investigated the humoral and T cell responses after primary BNT162b2 vaccination in secondary immunocompromised and healthy children aged 5 – 11 years. Overall, the topic is interesting and the manuscript is well-written. However, the case number is too limited to be presented as a full article. Therefore, I would recommend that the article could be shortened and accepted as a “Brief report”.
We thank the reviewer for this positive feedback and the valid suggestion. The aim of this study was to provide data in a homogenous and well characterized cohort of patients making the results applicable to a large number of children under immunosuppressive therapies including patients after solid organ transplantation. Considering the rare diseases studied, the case number is comparable to previously published manuscripts, especially focusing on the T cell response.
Additionally, the total number of patients who were recruited to achieve this overall homogenous cohort of patients that could finally be included in the analysis was much higher. In total 180 patients were initially recruited for this study. The selection process was conducted as follows: The vast majority of patients had to be excluded due to a natural SARS-CoV2 infection before or during the study period (infection before T2). Recruiting took place during an intense SARS-CoV-2 omicron infection wave. The second reason is a very common problem in pediatric studies, especially in rare diseases. The underlying diseases and schemes of immunosuppression had a high diversity in the total cohort so that often patients could not be grouped together and a statistically evaluable group size was not reached for all primary disease subgroups.
Due to the above-mentioned reasons, from our point of view, a Brief Report would not reflect the overall effort needed to conduct this study and prepare this manuscript and we therefore would like to keep the manuscript submitted as an article.
To simplify the manuscript and guide the reader’s attention on the achieved results we initially refrained from explaining the total recruiting process. A further explanation of the selection process was now added to the manuscript.
- 203 – 209: “A total of 180 participants were recruited for this study. Children who acquired a natural infection prior to T2 were excluded. Additionally, the recruited patients were characterized by a high diversity in the underlying diseases and schemes of immunosuppression. Thus, if a statistically evaluable group size was not reached for the respective primary disease subgroups those patients were also excluded from further analysis. The final cohort consisted of a total of 21 participants. “

Reviewer 2 Report
This is a quite interesting and well-designed and documented research.
Minor comments:
1. Figure 1: The rationale and strategy do belong to “Materials and Methods”, but the figure would rather move to “Results” together with the findings included in the legend.
2. L. 98-99: Probable natural infections with SARS-CoV-2 after T2 and before T3 might have functioned as additional immunisation events interfering with the T3 findings. Why is this not taken into account?
3. L. 244: “Specific responses” instead of “A specific response”.
4. L. 239, 278, 309 and 344: No vertical lines are seen on Figures 2, 3 a+b, 4 and 5. Should it be “horizontal”?
5. L. 285: “3.4 CD8+ T cell response” should be the title of the next unit.
6. L. 301: Yes, a specific CD8+ T cell response is also induced but, since there is no statistical significance, the fact that the response is only moderately induced should be clearly written.
Author Response
Point to point response to Reviewer 2
This is a quite interesting and well-designed and documented research.
Minor comments:
- Figure 1: The rationale and strategy do belong to “Materials and Methods”, but the figure would rather move to “Results” together with the findings included in the legend.
We thank the reviewer for this suggestion. We changed the order of the figures accordingly.
- 98-99: Probable natural infections with SARS-CoV-2 after T2 and before T3 might have functioned as additional immunization events interfering with the T3 findings. Why is this not taken into account?
We thank the reviewer for guiding or attention to that point. This was not made clear enough in our manuscript.
At each timepoint antibodies against the spike protein and nucleocapsid were measured to assess the possibility of an unrecognized natural infection. In only vaccinated participants, SARS-CoV-2 nucleocapsid antibodies would remain negative. At T2 only children that had no antibodies against nucleocapsid were considered. At T3 children that either had a natural infection (positive for nucleocapsid antibodies) or that were nucleocapsid negative and had a third vaccination were included to ensure that no additional immunization took place. An explanation was added to the methods part of the manuscript
L 105 – 107: “At T3 SARS-CoV-2 nucleocapsid IgG/M/A status was assessed. If a natural infection was acquired and a third vaccine dose was applied, patients were also excluded from timepoint T3 analysis.“
- 244: “Specific responses” instead of “A specific response”.
We thank the reviewer for this suggestion. The phrase was adapted accordingly.
- 239, 278, 309 and 344: No vertical lines are seen on Figures 2, 3 a+b, 4 and 5. Should it be “horizontal”?
We agree with the comment and changed the explanation to “horizontal”
- 285: “3.4 CD8+ T cell response” should be the title of the next unit.
We thank the reviewer for this hint. There was a formatting error and the heading was included in the figure caption above. We removed this mistake.
- 301: Yes, a specific CD8+ T cell response is also induced but, since there is no statistical significance, the fact that the response is only moderately induced should be clearly written.
We agree with the suggestion made by the reviewer. The paragraph in the results section was adapted accordingly.
- 315-319:” This data indicates that primary vaccination moderately enhances the development of a SARS-CoV-2 specific CD8+ T cell response to spike protein of Wuhan-Hu-1 and Omicron BA.5 strain. No significant difference in CD8+ T cell response, defined by SI, between healthy and immunocompromised patients could be seen. Indicating an inferior effect of vaccination on the CD8+ - T cell response, compared to the CD4+ T – cell response.“
